# Radical SAM-dependent ether crosslink in daropeptide biosynthesis

Sijia Guo[1,3], Shu Wang[2,3], Suze Ma[2], Zixin Deng [1], Wei Ding[1✉] & Qi Zhang [2✉]

Darobactin is a ribosomally synthesized and post-translationally modified peptide (RiPP), which possesses potent activity against various Gram-negative bacteria. Darobactin features a highly unique bicyclic scaffold, consisting of an ether crosslink between two Trp residues and a C–C crosslink between a Lys and a Trp. Here we report in vivo and in vitro activity of darobactin synthase DarE. We show DarE is a radical S-adenosylmethionine (rSAM) enzyme and is solely responsible for forming the bicyclic scaffold of darobactin. DarE mainly produced the ether-crosslinked product in vitro, and when the assay was performed in $H_2{}^{18}O$, apparent $^{18}O$ incorporation was observed into the ether-crosslinked product. These observations suggested an rSAM-dependent process in darobactin biosynthesis, involving a highly unusual oxygen insertion step from a water molecule and subsequent O–H and C–H activations. Genome mining analysis demonstrates the diversity of darobactin-like biosynthetic gene clusters, a subclade of which likely encode monocyclic products with only an ether linkage. We propose the name daropeptide for this growing family of ether-containing RiPPs produced by DarE enzymes.

[1] State Key Laboratory of Microbial Metabolism, School of Life Sciences & Biotechnology, Shanghai Jiao Tong University, Shanghai 200240, China. [2] Department of Chemistry, Fudan University, Shanghai 200433, China. [3]These authors contributed equally: Sijia Guo, Shu Wang. ✉email: weiding@sjtu.edu.cn; qizhang@sioc.ac.cn

The emergence of antimicrobial resistance is one of the major threats to global public health. This situation is especially troublesome with respect to pathogenic Gram-negative species, such as *Acinetobacter baumannii*, *Klebsiella pneumoniae*, *Pseudomonas aeruginosa* and *Enterobacter spp.* These Gram-negative bacteria are intrinsically resistant to many antibiotics, largely because of the low-permeability barrier resulting from the densely packed lipopolysaccharides in outer membrane[1]. Discovery of novel antibiotics to act against Gram-negative bacteria is challenging, and most frontline antibiotics (e.g., aminoglycosides, tetracyclines, β-lactams, fluoroquinolones) were introduced half a century ago. Due to the increased knowledge of bacterial genomes and the biosynthetic gene clusters (BGCs) of natural products, mining untapped groups of bacteria could perhaps be a fruitful strategy in finding antibiotics with novel scaffolds and modes of action[2,3].

Darobactin is a heptapeptide antibiotic found by an activity-based screening effort on nematophilic bacteria[4]. Darobactin features a unique bicyclic scaffold, comprising an ether linkage (C–O–C) between the C7 of $W^1$ and the $C_\beta$ of $W^3$, and a C–C crosslink between the C6 of $W^3$ and the $C_\beta$ of $K^5$ (Fig. 1a). Darobactin targets the bacterial insertase BamA, a central unit of the β-barrel assembly machinery (BAM) complex that is essential for the folding and insertion of outer membrane proteins[5]. The bicyclic scaffold of darobactin adopts a rigid β-strand conformation, thereby allowing the molecule to bind the lateral gate of BamA with an affinity in submicromolar range[6]. Darobactin exhibits potent activity against various multi-drug resistant Gram-negative bacteria both in vitro and in animal models. Moreover, it shows no activity against common symbiotic gut bacteria and human cell lines, presenting as a promising lead compound for developing therapeutics against Gram-negative pathogens[4].

Darobactin belongs to the ribosomally synthesized and post-translationally modified peptide (RiPP) superfamily[7,8]. The darobactin BGC encodes a 58-amino acid (aa) precursor peptide DarA, a radical S-adenosylmethionine (rSAM) enzyme DarE, and an ABC-type transporter DarBCD (DarB and DarD are the transporters whereas DarC is a membrane fusion protein) (Fig. 1b)[4]. The rSAM superfamily enzymes are extensively involved in RiPP biosynthesis and catalyze strikingly diverse reactions[9–11]. The rSAM-dependent cyclic RiPPs include sactipeptide (α-thioether linkage)[12–18], ranthipeptide (β or γ-thioether linkage)[19–21], lytreptides (C–C crosslink between Lys and Trp)[22,23], rotapeptide (C–O crosslink between Thr and Gln)[24], ryptide (C–C crosslink between Arg and Tyr)[25], triceptide (C–C crosslink between an aromatic aa and a non-aromatic aa)[26,27], among others (Supplementary Fig. 1). However, the bicyclic scaffold of darobactin, particularly the ether crosslink between two Trp residues, is distinct from all the known rSAM-dependent RiPPs. Recent heterologous expression studies showed that the minimum darobactin BGC consists of only *darA* and *darE*, whereas *darBCD* is not essential for darobactin production[27–29]. However, the pathway for darobactin maturation and the exact function of DarE remains unclear.

Herein, we show DarE is solely responsible for the bicyclic scaffold formation in darobactin biosynthesis. We also show darobactin analogs are widespread in nature with high structural diversity, and we propose the name daropeptide for this growing family of RiPPs.

## Results and discussion

**In vivo activity of $DarE_{pk}$.** To investigate darobactin biosynthesis, we coexpressed *darE* from *Photorhabdus khanii* HGB1456 with its precursor peptide gene *darA* in *Escherichia coli* (the two genes are referred to as $darE_{pk}$ and $darA_{pk}$ to differentiate from the homologs from other strains). In contrast to the recent coexpression studies that focused on darobactin production[28,29], we expressed $darA_{pk}$ with an N-terminal hexa-histidine tag (Supplementary Fig. 2) and purified the peptide by $Ni^{2+}$-affinity chromatography. Liquid chromatography with high-resolution mass spectrometry (LC-HRMS) analysis showed the resulting $DarA_{pk}$ exhibited a protonated molecular ion at $[M + 8H]^{8+} = 1011.0$, which is 12 Da higher than the expected unmodified DarA (hereafter the fully modified +12 Da peptide is termed $DarA_{pk}$-**1**) (Fig. 2a). $DarA_{pk}$-**1** was then digested by trypsin and analyzed by high-resolution MS (HR-MS) (Fig. 2a) and tandem MS (HR-MS/MS) (Supplementary Fig. 3). This analysis clearly revealed that the tryptic fragment $DarA_{pk(43-58)}$-**1** ($[M + 2H]^{2+}$ cal. 980.9756, obs. 980.9756, err < 0.1 ppm) contains the expected crosslink between Trp49 and Lys53 (Fig. 1a), suggesting that the DarE enzyme is responsible for installing the characteristic bicyclic scaffold of darobactin on DarA. Further introduction of $darBCD_{pk}$ into the coexpression system (to avoid any possible issues in relation to gene regulation, all the genes were each expressed under the control of a separate T7-lac promoter, Supplementary Fig. 2) did not have an observable effect in the yield and modification pattern of $DarA_{pk}$ (Supplementary Fig. 4). This observation suggests that DarBCD do not play an essential role in darobactin biosynthesis, which is consistent with the recent heterologous studies[28–30].

The rSAM superfamily enzymes utilize a [4Fe-4S] cluster to reductively cleave S-adenosylmethionine (SAM) to generate an extremely reactive 5'-deoxyadenosyl (dAdo) radical, which initiate highly diverse reactions[31,32]. This [4Fe-4S] cluster is bound by three Cys residues, which are usually organized in a CxxxCxxC motif. In our analysis, we changed Cys83 in the CxxxCxxC motif of $DarE_{pk}$ (i.e., CNLRCTYC) to an Ala, and the mutant gene was coexpressed with $darA_{pk}$, using the same protocol as that for the wild type $darE_{pk}$. Subsequent LC-HRMS analysis showed that the expected unmodified $DarA_{pk}$ was not observed in the purified peptide sample. Instead, the resulting peptide is a truncated peptide $DarA_{pk(1-48)}$ that only contains the leader region of $DarA_{pk}$, whereas the core and follower peptide regions (i.e., $W_{49}$NWSKSFQEI$_{58}$) were removed from the precursor peptide (Fig. 2b). This observation suggests that the

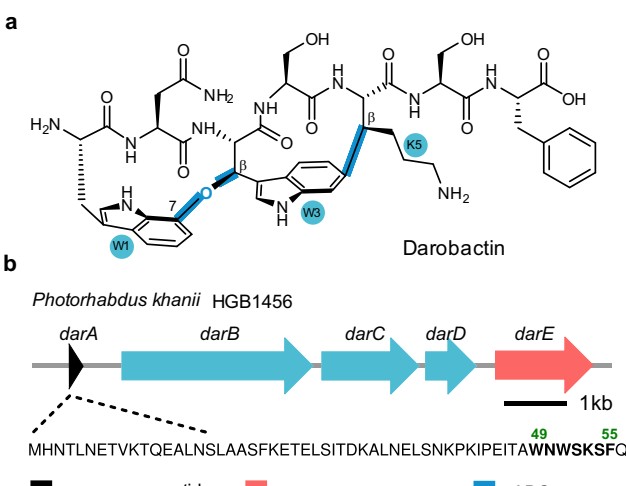

**a**

Darobactin

**b**

*Photorhabdus khanii* HGB1456

*darA*  *darB*  *darC*  *darD*  *darE*

1kb

MHNTLNETVKTQEALNSLAASFKETELSITDKALNELSNKPKIPEITA**WNWSKSF**QEI

■ precursor peptide  ■ radical SAM enzyme  ■ ABC transporter

**Fig. 1 Darobactin and its BGC. a** The chemical structure of darobactin. The key C–O–C and C–C crosslinks that consist the unique bicyclic scaffold of darobactin are highlighted in blue. **b** The darobactin BGC from *Photorhabdus khanii* HGB1456 encodes a precursor peptide $DarA_{pk}$, an rSAM enzyme $DarE_{pk}$, and three ABC transports $DarBCD_{pk}$. The core sequence of $DarA_{pk}$ ($DarA_{pk(49-55)}$) is shown in bold.

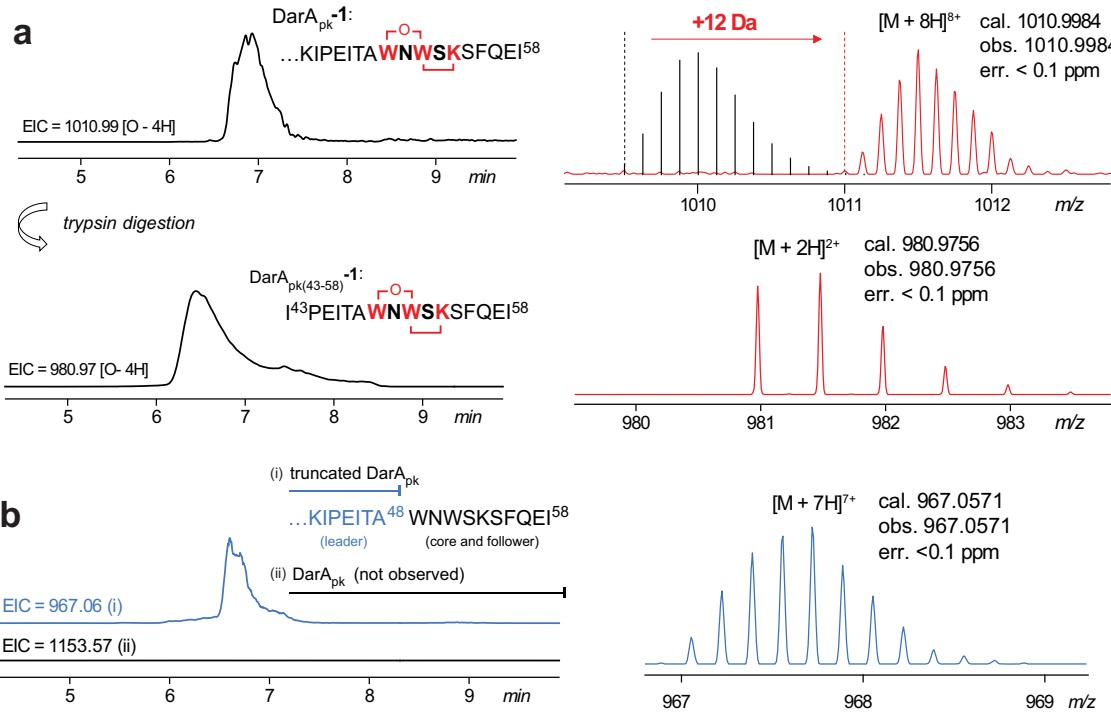

**Fig. 2 LC-HRMS analysis of DarA$_{pk}$ produced in vivo, showing the extracted ion chromatograms (EICs) and MS spectra. a** DarA$_{pk}$ produced by coexpression with DarE$_{pk}$ carries a + 12 Da modification. Because heterologous expression in *E. coli* only produced obtained the truncated DarA$_{pk}$, the theoretically predicted signals corresponding to the unmodified DarA$_{pk}$ were shown for comparison. See Supplementary Fig. 3 for the HR-MS/MS spectrum of DarA$_{pk(43-58)}$-**1**. **b** DarA$_{pk}$ produced by coexpression with the C86A mutant of DarE$_{pk}$ is a truncated peptide (i.e., DarA$_{pk(1-48)}$), which lacks the core and follower regions.

leader peptide of DarA$_{pk}$ is naturally cleaved by one or more unknown hydrolases in the proteome of *E. coli* (as well as the native host *P. khanii*). Formation of the bicyclic scaffold of darobactin likely decreases the proteolysis efficiency, allowing for the purification of the full-length modified DarA$_{pk}$-**1** in our analysis (Fig. 2a). Similar to the recent reports[28,29], production of the mature darobactin is also observed in our analysis (Supplementary Fig. 5).

To show that cleavage of DarA leader and follower peptide was achieved by *E. coli* proteins, we incubated the in vivo modified DarA$_{pk}$-**1** with the cell lysate of *E. coli* BL21. In this analysis, we saw apparent production of darobactin in the reaction (Supplementary Fig. 6). This result clearly demonstrates that proteolytic cleavage of DarA for darobactin maturation is catalyzed by one or more unknown hydrolases in the *E. coli* proteome.

**In vitro analysis of DarE$_{pk}$.** We next set out to investigate the in vitro activity of DarE$_{pk}$. To this end, we overexpressed DarE$_{pk}$ in *E. coli* with an N-terminal hexa-histidine tag. However, despite extensive efforts the protein was mainly seen in the inclusion bodies. Because in our coexpression study DarE$_{pk}$ is apparently functional and can efficiently install the bicyclic scaffold on DarA$_{pk}$, we reasoned the presence of the precursor peptide could likely assist the folding of DarE$_{pk}$. Indeed, coexpression of the untagged DarA$_{pk}$ with the N-terminally His-tagged DarE$_{pk}$ significantly increased protein solubility. By this approach ~30 mg DarE$_{pk}$ was obtained from 2 L cell culture by Ni$^{2+}$-affinity chromatography under strictly anaerobic conditions. The as-isolated protein appeared brownish and exhibited a weak broad absorption around 410 nm, and this absorption band significantly increased after chemical reconstitution of the [4Fe-4S] cluster (Supplementary Fig. 7). Quantification analysis showed that each

enzyme contains 7.8 ± 0.5 iron and 7.2 ± 0.6 labile sulfide, suggesting it harbors two [4Fe-4S] clusters. This observation is consistent with the fact that DarE contains a C-terminal SPASM/twitch domain, which binds additional [4Fe-4S] clusters, with roles suggestive of peptide binding or electron transfer[33,34]. Anaerobic incubation of the reconstituted DarE$_{pk}$ with SAM and sodium dithionite (DTH) produced a significant amount of 5′-deoxyadenosine (dAdoH) (Supplementary Fig. 7), validating that DarE$_{pk}$ is an rSAM enzyme.

Similar to the above-mentioned efforts in coexpression of DarA$_{pk}$ with the C83A mutant of DarE$_{pk}$, expression of DarA$_{pk}$ alone in *E. coli* only afford the truncated peptide DarA$_{pk(1-48)}$. We hence obtained the full-length DarA$_{pk}$ by solid-phase peptide synthesis (SPPS) (Supporting Methods), and the assay was then performed by treating DarA$_{pk}$ with the reconstituted DarE$_{pk}$, SAM, and DTH under strictly anaerobic condition. The reaction mixture was then treated with trypsin and analyzed by LC-HRMS. This analysis revealed an apparent product corresponding to the tryptic fragment DarA$_{pk(43-53)}$ (i.e., I$_{43}$PEITAWNWSK$_{53}$) harboring +14 Da modification ([M + 2H]$^{2+}$ cal. 679.8406, obs. 679.8505, err. 0.1 ppm), and this product was clearly absent in the control assays with the supernatant of boiled enzyme (Fig. 3a). The +14 Da tryptic product (hereafter referred to as DarA$_{pk(43-53)}$-**2**) is consistent with the modified DarA$_{pk}$ containing an ether crosslink between Trp49 and Trp51, which is further supported by HR-MSMS analysis (Supplementary Fig. 8).

We also observed a trace amount of fully modified DarA$_{pk}$ harboring the +12 Da modification (i.e., DarA$_{pk}$-**1**), whose tryptic fragment DarA$_{pk(43-58)}$-**1** exhibit a protonated molecular ion at [M + 2H]$^{2+}$ = 980.9756 (cal. 980.9756, err. <0.1 ppm) (Fig. 3b). Although the in vitro reaction efficiency of DarE$_{pk}$ is low and needs further optimization, these results clearly demonstrate that DarE installs the bicyclic scaffold of darobactin on DarA via

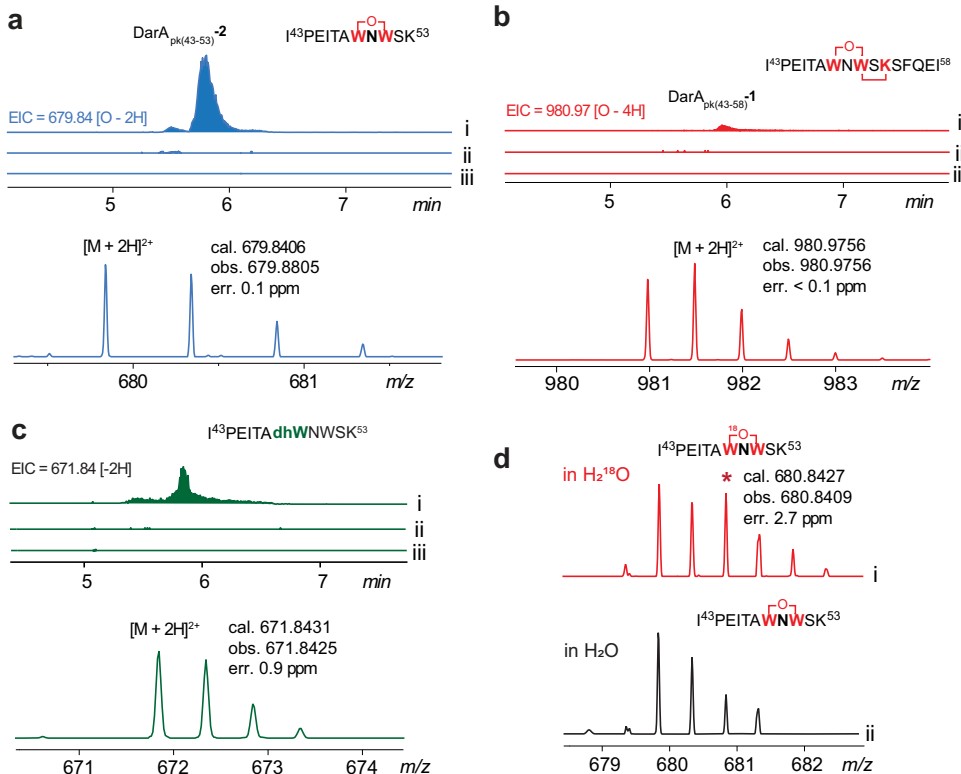

**Fig. 3 In vitro analysis of DarE$_{pk}$. a** The EIC and HRMS spectrum corresponding to the tryptic fragment DarA$_{pk(43-53)}$-**2**, which contains an ether-crosslink between Trp49 and Trp51. **b** The EIC and HRMS spectrum corresponding to the tryptic fragment DarA$_{pk(43-53)}$-**1** containing the fully modified bicyclic scaffold. **c** The EIC and HRMS spectrum corresponding to the tryptic fragment DarA$_{pk(43-53)}$-**3** likely containing a dehydrogenated Trp51 (dhW). Trace i, ii, and iii correspond to the full reaction, the solution of reconstituted DarE$_{pk}$, and the negative control with the supernatant of heat-inactivated enzyme. **d** HRMS spectra of the tryptic fragments DarA$_{pk(43-53)}$-**2** generated in (i) a buffer containing ~80% H$_2$$^{18}$O, which was obtained by ultrafiltration and H$_2$$^{18}$O dilution. The asterisk indicates $^{18}$O incorporation (~50% enrichment) into the resulting peptide. A control experiment (ii) was performed in parallel with H$_2$O. See Supplementary Fig. 12 for the detailed comparison of the HR-MSMS spectra for the $^{18}$O-labeled vs. unlabeled tryptic fragments DarA$_{pk(43-53)}$−**2**.

rSAM-dependent chemistry. We did not observe the product corresponding to the peptide only containing the Trp51-Lys53 crosslink (i.e., having a C–C crosslink but lacking the ether linkage). Together with the fact that the ether-crosslinked DarA$_{pk}$-**2** is the major product in the assay, this observation suggests that ether bond formation between Trp49 and Trp51 is likely the first step in darobactin biosynthesis. We also constructed a K53A mutant of DarA$_{pk}$ in which the to-be-cyclized Lys53 was changed to Ala, and the mutant was coexpressed with DarE$_{pk}$. LC-HRMS and HR-MSMS analysis clearly indicated the resulting DarA$_{pk}$ mutant mainly harbored a + 14 modication, which occurred between Trp48 and Trp51 (Supplementary Fig. 9). This result is consistent with the proposal that ether bond formation precedes the C–C bond formation in darobactin biosynthesis.

Interestingly, careful analysis of the in vitro reaction mixture revealed a tryptic fragment DarA$_{pk(43-53)}$ harboring -2 Da modification ([M + 2H]$^{2+}$ cal. 671.8431, obs. 671.8425, err. 0.9 ppm), and this product (hereafter referred to as DarA$_{pk(43-53)}$-**3**) was clearly absent in the control assays with the supernatant of boiled enzyme (Fig. 3c). This −2 Da product is apparently not related to the Trp51-Lys53 crosslink (i.e., the C–C crosslink between W$^3$ and K$^5$ in darobactin), which would preclude tryptic cleavage between Lys53 and Ser54. Detailed HR-MS/MS analysis showed the −2 Da modification occurred on Trp51 (Supplementary Fig. 10), suggesting the production of dehydrogenated Trp (dhW) in DarE reaction. Dehydrogenation of Trp51 raises an interesting possibility that the β-hydroxyl moiety of Trp in darobactin is likely a result dhW hydration. To test this

hypothesis, we performed the assay in a buffer containing ~80% H$_2$$^{18}$O, and the reaction mixture was examined by HR-LCMS. This analysis clearly revealed $^{18}$O incorporation into the corresponding DarA$_{pk(43-53)}$-**2** fragment (Fig. 3d). Detailed HR-MSMS analysis showed that the $^{18}$O atom is incorporated between Trp49 and Trp51 (Supplementary Figs. 11–12), suggesting the oxygen atom in the ether crosslink of darobactin is derived from H$_2$O.

Based on the results presented above, we proposed a working hypothesis for the DarE-catalyzed reaction (Fig. 4). In the reaction, the dAdo radical generated from SAM cleavage abstracts a hydrogen from the side chain of Trp51, leading to the production of the dhW-containing product DarA-**3**. Because DarE shares apparent sequence similarity with tricepeptide synthases (e.g., 38% similarity with SjiB[27]), this hydrogen abstraction process very likely occurs on the β-carbon of Trp51 (Fig. 4). DarA-**3** is subsequently hydrated via a stereoselective Michael addition to produce DarA-**4**. The dAdo radical produced from the second SAM molecules then abstracts the hydrogen from the β-hydroxyl group of DarA-**4** to produce an oxygen-centered radical **X**, which then attacks the C7 indole ring of Trp49 to install the C–O–C linkage (Fig. 4), in a way similar to that proposed in glycopeptide biosynthesis[35,36]. It is noteworthy that although hydrogen abstraction from an alcohol hydroxyl group is rare in biochemistry, this type of chemistry has been demonstrated in NosL catalysis with unnatural substrates[37–39]. After the formation of the ether crosslink, the dAdo radical from the third SAM molecule then installs the C–C crosslink between Lys53 and Trp49; similar type of chemistry has been well

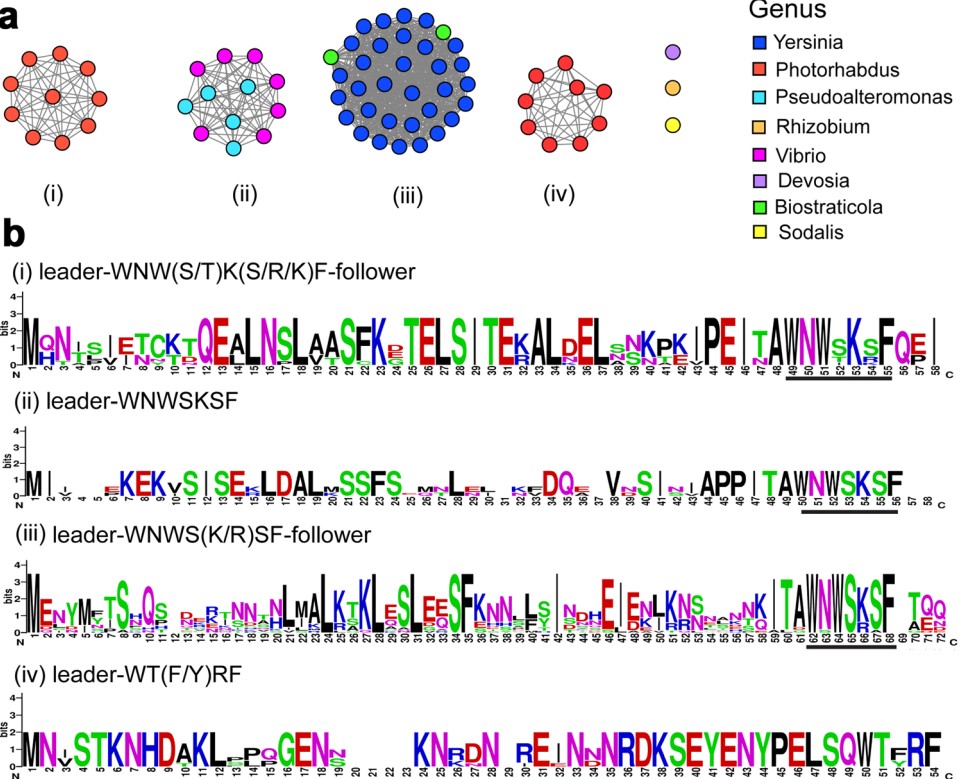

**Fig. 4 A proposed mechanism for the DarE-catalyzed bicyclic crosslinking.** N in the circle represents the Asn residue between the two Trp residues in DarA (e.g., $N^2$ in darobactin).

**Fig. 5 Diversity and classification of darobactin-like BGCs. a** The sequence similarity network of DarE enzymes. Each node in the network represents a putative DarE sequence, and each edge represents sequences with BLASTP E values below 1E-165. **b** The conserved core sequences in the putative precursor peptides are shown by black bars.

characterized in various rSAM-dependent RiPPs such as streptide[22,23]. Owing to the complex chemistry of DarE, further investigations are awaited to validate the detailed mechanism of this remarkable catalytic process.

**The substrate diversity of DarE enzymes.** Recent heterologous studies demonstrated the great potential to expand the structural diversity of daropeptide by bio-engineering efforts[28,29]. To explore the catalytic protential and substrate specificity of DarE enzymes, we carried out a BLASTp search in UniProt database using DarE$_{pk}$ as query sequence. The putative darobactin synthases were then selected by analyzing the neighboring sequences

of these enzymes with the assistance of the EFI-genome neighborhood tool[40]. The identified enzymes were then incorporated into a sequence similarity network (Fig. 5a), and the corresponding precursor peptides were analyzed by multiple sequence alignment (Fig. 5b).

In this analysis, putative DarE enzymes appear to fall into four clusters (Fig. 5a), and the corresponding precursor peptides in each cluster also contain different core sequences (Fig. 5b). The cluster I consists of sequences from *Photorhabdus*, which includes the known darobactin synthase DarE$_{pk}$. The precursor peptides from cluster I contain the conserved WNWxKxF core peptide, representing the typical bicyclic scaffold of darobactin. Cluster II consists of sequences from diverse genera, and the core peptides

WNWSKSF are similar to those from cluster I, but these precursor peptides mostly do not have a follower sequence (e.g., the last two aa in DarA$_{pk}$). Cluster III mainly consists of sequences from *Yersinia*, and the precursor peptides contain a WNWx(K/R)xF core, suggesting the crosslinked Lys could be replaced by an Arg. Like cluster I, the sequences from cluster IV are also from *Photorhabdus*; however, the precursor peptide sequences of cluster IV (WT(F/Y)RF) are apparently distinct from those of the other three clusters.

To interrogate the DarE-catalyzed modification on other precursor peptides, we first focused on a cluster III BGC from *Yersinia intermedia*, which encodes a precursor peptide (hereafter DarA$_{yi}$) containing an Arg substitution in the core (i.e., WNWS<u>R</u>SF). Similar to the coexpression study discussed above, we coexpressed DarA$_{yi}$ in an N-terminal hexa-histidine tagged form with the corresponding modifying enzyme DarE$_{yi}$, and the resulting peptide was purified by Ni$^{2+}$-affinity chromatography. LC-HRMS analysis revealed an apparent peak of +12 Da relative to the unmodified DarA$_{yi}$ (Fig. 5a) consistent with the formation of the expected bicyclic scaffold. Upon trypsin digestion followed by LC-HRMS analysis, we observed a peptide fragment exhibiting a protonated molecular ion at $[M + 2H]^{2+} = 740.3349$ (cal. 740.3362, err. 1.7 ppm), which corresponds to the fully modified C-terminus of DarA$_{yi}$ (i.e., DarA$_{yi(55-66)}$-**1**) (Fig. 6a), and the crosslink between Trp58 and Arg62 is further supported by HR-MS/MS analysis (Supplementary Fig. 13).

To validate that the bicyclic scaffold produced on DarA$_{yi}$ has the same regio- and stereo-chemistry as that of darobactin, we set out to characterize the tryptic fragment DarA$_{yi(55-66)}$-**1** by NMR. To this end, we performed coexpression experiment in a large scale, and the fully modified DarA$_{yi}$-**1** was purified from 60 L cell culture and was then treated with trypsin. Multiple rounds of semi-preparative LC, guided by the LC-MS, afford ~1 mg purified DarA$_{yi(55-66)}$-**1**. $^1$H NMR analysis of the resulting compound revealed several exchangeable amide NH protons ($\delta_H$ 5.6–8.4

ppm), confirming it is a peptide (Supplementary Fig. 14). Although a high degree of signal overlap in the $^1$H NMR spectrum, twelve spin systems were assembled from the integrative analysis of 2D NMR data, including COSY, HSQC, and NOESY (Fig. 6b, Supplementary Figs. 15–17 and Supplementary Table 1), which are in full agreement with the expected DarA$_{yi(55-66)}$ sequence. Importantly, a significant downfield shift of a doublet at 5.99 ppm (W$^4$-H$\beta$) clearly indicates oxidation of the W4 C$\beta$, which appear as a methine at 75.19 ppm. By a combination of vicinal coupling ($^3$J), NOESY correlations, and biogenetic considerations, the newly generated two stereocenters at the C$\beta$ of W$^6$ and R$^8$ were assigned to be *R*- and *S*-configured, respectively. These results revealed the bicyclic scaffold on DarA$_{yi(55-66)}$ is identical to that reported for the darobactins (Fig. 1)[4,29]. We also performed the density functional theory (DFT) calculation using the gauge-including atomic orbital (GIAO) method[41,42], and the calculation is in good agreement with the experimental data (Supplementary Fig. 18). Together, our analysis indicates that DarE installs the characteristic bicyclic scaffold of darobactin on a Lys-to-Arg variant of DarA with the same regio- and stereo-selectivity.

We next focused a cluster IV BGC from *Photorhabdus asymbiotica* (Fig. 6a). To this end, the precursor peptide DarA$_{pa}$ was coexpressed with the modifying enzyme DarE$_{pa}$, and the resulting purified peptide was analyzed by LC-HRMS analysis. This analysis showed the resulting DarA$_{pa}$ (i.e., DarA-**1**) harbors +14 Da modification ($[M + 8H]^{8+}$ obs. 929.5505, cal. 929.5506), suggesting the formation of an ether crosslink (Supplementary Fig. 19). LC-HRMS analysis of the trypsin-digested sample revealed a peptide fragment (i.e., DarA$_{pa(32-49)}$-**1**) exhibiting a protonated molecular ion at $[M + H]^{2+} = 1177.0314$ (cal. 1177.0334, err. 1.7 ppm) (Supplementary Fig. 20). Subsequent HR-MS/MS analysis clearly revealed the ether crosslink in DarA$_{pa(32-49)}$-**1** is formed between Trp45 and Phe47 (Supplementary Fig. 20). This analysis suggested that the class IV BGCs

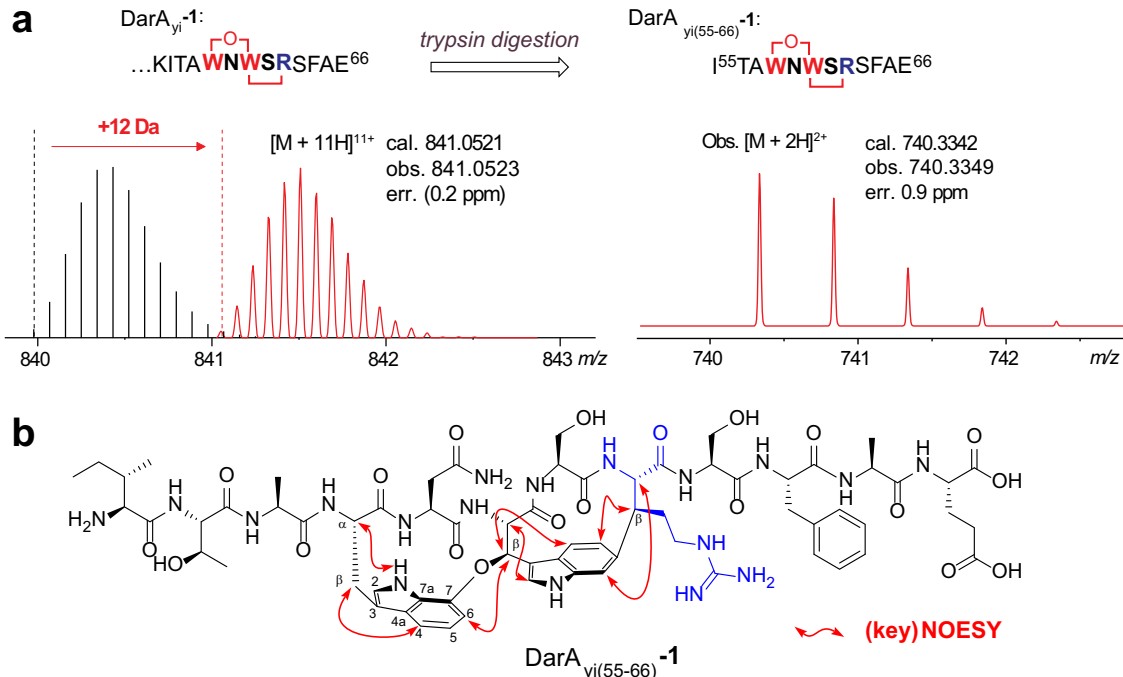

**Fig. 6 An Arg-containing daropeptide from *Yersinia intermedia*. a** HRMS analysis of the fully modified DarA$_{yi}$ (DarA$_{yi}$-**1**) produced by coexpression of DarA$_{yi}$ with DarE$_{yi}$. Similar to Fig. 1, the theoretically predicted signals corresponding to the unmodified DarA$_{yi}$ were shown for comparison. **b** The chemical structure of DarA$_{yi(55-66)}$-**1**, showing the key NOSEY correlation. See Supplementary Figs. 10–13 for the 1D and 2D NMR spectra and Supporting methods for details.

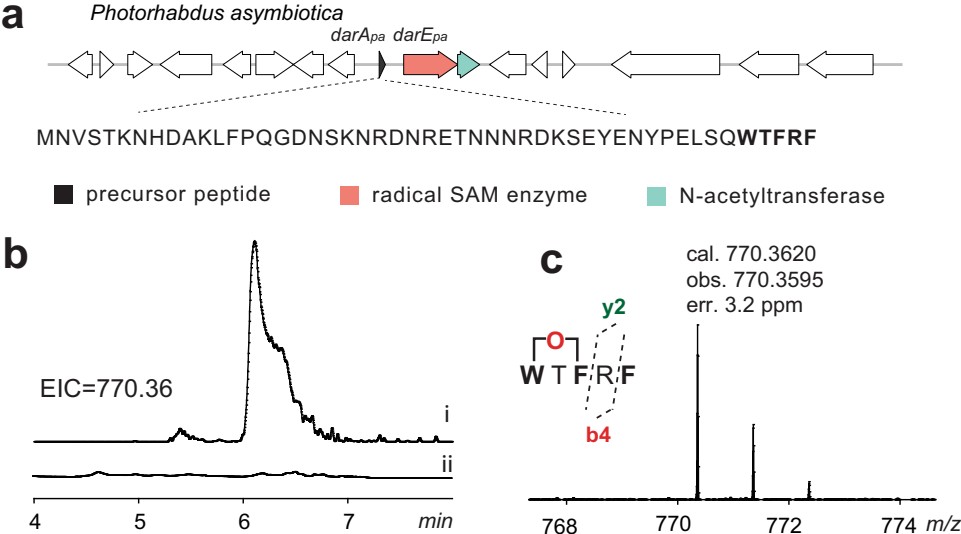

**Fig. 7 Production of a monocyclic daropeptide from a cluster IV BGC. a** The BGC from *P. asymbiotic*, showing the genes and the precursor peptide sequence, and the core sequence is shown in bold font. Unlike other daropeptide BGCs, this BGC does not encode the DarBCD transporters but encodes a putative N-acetyltransferase. **b** The EIC at $m/z = 770.36 \pm 0.01$ (corresponding to the monocyclic pentapeptide containing an ether crosslink) of DarA$_{pa}$-**1** incubated with (i) the cell lysate of *E. coli* BL21 or (ii) the boiled cell lysate for 24 h. **c** HR-MS spectrum of the in vitro obtained monocyclic daropeptide. See Supplementary Figs. 17–19 for the data in characterization of the DarA$_{pa}$-derived peptides.

likely encode monocyclic peptides that only contain an ether crosslink. To further test this hypothesis, we performed the in vitro proteolytic assay by treating the in vivo obtained DarA$_{pa}$-**1** with the cell lysate of *E. coli* BL21, similar to the in vitro darobactin production discussed above. Indeed, this analysis clearly revealed the production of a compoud exhibiting a protonated molecular ion at $[M + H]^+ = 770.3595$ (cal. 770.3620, err. 3.2 ppm) (Figs. 7b and 7b). This compound corresponds to a pentapeptide WTFRF containing an ether crosslink between $W^1$ and $F^3$, and this proposal is further supported by HRMS/MS analysis (Supplementary Fig. 21). Although we failed to obtain enough amount of peptide for NMR characterization owing to its very low yield, we expect the ether crosslink of DarA$_{pa}$-**1** is formed with the same regio- and stereo-chemistry as that of darobactin. Future studies are awaited to further characterize these monocyclic darobactin-like natural products encoded by the class IV BGCs.

The DarE$_{pa}$-catalyzed modification on DarA$_{pa}$ suggested that the ether-crosslink can be not only formed between two Trp residues, but also between a Trp and a Phe. To test this hypothesis, we constructed the W51F mutant of DarA$_{pk}$, in which the following Trp (i.e., W3 in the final product) was changed to a Phe, and the mutant was coexpressed with DarE$_{pk}$. LC-HRMS and HR-MSMS analysis showed the resulting DarA$_{pk}$ mutant mainly harbored a $+14$ Da modication, which occurred between Trp48 and Phe51 (Supplementary Fig. 22). This result suggested the ability of ether-crosslink between a preceding Trp and a following Phe is likely common to enzymes from all the clusters. We also constructed the W49F mutant of DarA$_{pk}$, in which the preceding Trp (i.e., W1 in the final product) was changed to a Phe. LC-HRMS and HR-MSMS analysis showed the resulting DarA$_{pk}$ mutant coexpressed with DarE$_{pk}$ harbored a $+16$ Da modification on Trp51, and the ether-crosslink was not formed (Supplementary Fig. 23), suggesting that the enzymes appeared unable to form ether-crosslink between a preceding Phe and a following Trp. Observation of the hydroxylated Trp51 in this analysis is also consistent with the mechanisic proposal that W3 is first hydroxylated prior to ether crosslink formation (Fig. 4).

To test the enzymes of different clusters are promiscuous toward precursor peptides of other clusters, we coexpressed DarA$_{pk}$ with DarE$_{yi}$ and DarE$_{pa}$, respectively. In contrast to complete formation of the bicyclic strcutre observed for the naitve enzme DarE$_{pk}$, only ether-crosslink (i.e, $+14$ Da modification) was observed for the DarA$_{pk}$ peptide coexpressed with DarE$_{yi}$ and DarE$_{pa}$ (Supplementary Figs. 24–25). These results suggested that although DarA can be modified by DarE enzymes from different clusters, these enzymes are likely not as efficient as the naitve enzyme. Observation of the monocyclic ether-crosslinked DarA$_{pk}$ in these analysis again supported that ether-crosslink precedes the C–C bond formation in darobactin biosynthesis.

In summary, we show DarE is solely responsible for producing the bicyclic scaffold of darobactin. This remarkable double crosslinking reaction is dependent on rSAM chemistry, and formation of the ether link between two Trp residues precedes the formation of C–C crosslink of darobactin. Identification of a Trp-dehydrogenated product and isotopic labeling studies shed mechanistic insights into the rSAM-dependent ether crosslinkage, highlighting the remarkable catalytic repertoire and intriguing chemistry of the rSAM superfamily enzymes. Our study also demonstrates a unique RiPP biosynthetic pathway, involving an rSAM-dependent crosslinkage and proteolysis by utilizing the peptidases/hydrolases from the host proteomes.

Our study also demonstrates the rich reservoir of darobactin-like BGCs and the potential in investigating this family of RiPPs by genome mining and biosynthetic engineering efforts. Besides the bicyclic scaffold, monocyclic scaffolds with only an ether crosslink can also be produced, which could be the final products (i.e., clade IV BGC) or intermediates resulted from the inefficient enzyme catalysis. We propose the name daropeptides for this growing family of RiPPs that feature an aromatic-aliphatic ether crosslinkage produced by DarE enzymes. Future studies to explore the chemical space and biological activity of this intriguing family of RiPPs could be fruitful, particularly for the development of novel antibiotic lead compounds against Gram-negative pathogens.

## Methods

**Production of the DarE-modified DarA.** The corresponding coexpression plasmids containing both *darA* and *darE* genes were transformed into chemical competent BL21(DE3) cells by heat shock. Cells were grown for 16 h on LB agar plate at 37 °C. A single colony transformant was used to inoculate 10 mL of LB medium and grown at 37 °C overnight. This culture was used to inoculate 1 L of TB medium, incubated at 37 °C/200 rpm until an $OD_{600}$ of 0.6–0.8 was reached, and then induced with IPTG to a final concentration of 0.2 mM. The cultures were then incubated at 18 °C with for 18 h. The cells were harvested by centrifugation (4000 x g for 10 min), and were directly subjected to protein purification.

The cell paste was resuspended in about 20 mL denaturing buffer I (6 M guanidine hydrochloride, 20 mM Tris, 500 mM NaCl, 0.5 mM imidazole, pH = 7.5) and then lysed by sonication on ice. The sample was centrifuged at 12,000 x g for 40 min at 4 °C. The insoluble portion was discarded and the resulting supernatant was clarified through 0.45 mm syringe filters. The peptide-containing sample was the loaded to a 3 ml Ni-NTA column pre-equilibrated with buffer I. The resin was successively washed with 2 column volumes (CV) of buffer I and 4 CV of buffer II (4 M guanidine hydrochloride, 20 mM Tris, 300 mM NaCl, 30 mM imidazole, pH = 7.5), and the desired peptide was eluted using 3 CV of elution buffer (4 M guanidine hydrochloride, 20 mM Tris, 100 mM NaCl, 500 mM imidazole, pH = 7.5). The peptide was then desalted by semi-preparative HPLC using the YMC-Triart C8 column (10.0 × 250 mm, 5 mm). Mobile phase for the crude purification was made of solvent A (0.1% TFA in $H_2O$) and solvent B (MeCN). A linear gradient of 5–30% of solvent B over 30 min was executed with a flow rate of 3.0 mL/min. Target peptides were monitored by UV absorbance at 210 and 278 nm, and normally eluted around 20~22 min. Collected fractions were lyophilized and a white fluffy solid was obtained (typically 5 mg of final dried peptide per liter of overexpressed cells), which was further sent for LC-HRMS analysis. LC-HRMS in positive ion mode was operated at 0.3 mL/min with solvent A (0.1% formic acid in $H_2O$) and solvent B (MeCN) under the following condition: 0–2 min, 2%B; 2–5.5 min, 2–40%B; 5.5–7.5 min, 40%B; 7.5–9.5 min 95%B. Collision-induced dissociation was a method used for fragmentation, and normalized collision energy was 45 eV.

To analyze the proteolysis fragments of the modified substrate DarA, the lyophilized peptides were resuspended in a 100 mL of Tris buffer (20 mM Tris, 25 mM NaCl, pH 8.0) and incubated with trypsin (1:100, trypsin/peptide w/w) at 37 °C for 3 h. After boiling and centrifugation, the supernatant was directly injected to LC-HRMS analysis. The proteolytic digestion of DarA variants using trypsin followed the same procedure unless otherwise specified.

**In vitro activity assays for DarE_{pk}.** In vitro assays were usually measured in a 100 μL reaction volume containing DarE (40 μM), precursor peptide DarA (synthesis, 200 μM), sodium dithionite (5 mM), SAM (2 mM) and reaction buffer (50 mM MOPS, pH = 7.5). Boiled enzyme was provided for control assay. The reaction was allowed to process 8–10 h at 30 °C in an anaerobic chamber before quenching with 100 mL MeCN. The precipitant was removed by centrifugation at 13,000 rpm for 15 min and then concentrated volume by SpeedVac for 10 min. The supernatant was applied to be analyzed by LC-HRMS.

For the assay in $H_2^{18}O$, the reaction mixture was prepared similar to the normal assay with the omitting of SAM. The mixture was then subjected to two rounds of buffer exchange by ultrafiltration and $H_2^{18}O$ dilution to achieve a buffer containing ~80% $H_2^{18}O$. The assay was then initiated by addition of the SAM dissolved in $H_2^{18}O$. A parallel experiment performed with ultrafiltration and $H_2O$ dilution was also performed for comparative analysis.

Trypsin digestion of reaction mixture was performed in trypsin digestion buffer (50 mM Tris, 20 mM $CaCl_2$, pH = 7.5). 10 mL freshly prepared trypsin solution (2 mg/ml) was added to 20 mL reaction mixture and replenished the trypsin digestion buffer to a total volume of 50 mL. After incubation at 37 °C for 3 h, 5% (v/v) TFA was added to the solution to quench reaction, and the supernatant was analyzed by LC-HRMS.

**In vitro proteolysis.** To prepare the cell lysate, cell pellet of *E. coli* BL21(DE3) was resuspended in 20 mL of the lysis buffer, lysed by sonication on ice, and centrifuged at 14,000 rpm for 10 mL. 17 mL (1 mg/mL) modified precursor peptide was incubated with 83 mL supernatant of cell lysate. For the control assays, Dar-A_{pk}-**1** was incubated in lysis buffer or treated with the boiled cell lysate. The reaction mixtures were quenched by boiling for 5 min, the precipitated protein was removed by centrifugation, and the respective supernatants were subjected to LC-HRMS analysis.

**Genome mining.** To discover DarE homologs, the sequence similarity network (SSN) was constructed by EFI-Enzyme Similarity Tool (EST)[40] (https://efi.igb.illinois.edu/efi-est/) using the Option A 'sequence BLAST' with DarE_{pk} (WP_152942147.1) as query input under default parameters (which is used to BLAST in UniProt sequence database to retrieve 5000 sequences with a BLAST *E*-value at $10^{-5}$). The SSN containing 5000 DarE homologs was generated with an *E*-value cutoff at $10^{-170}$ (90% identical sequences are conflated) and visualized in Cytoscape[43] 3.4.0. using organic layout. The SSN was subsequently uploaded to EFI-genome neighborhood tool (GNT) (https://efi.igb.illinois.edu/efi-gnt/) to manually examine the genomic neighborhood of the 5000 DarE homologs so as to discover BGCs encoding potential

darobactin analogs. Multiple sequence alignment of the precursor peptide candidates was generated by using MAFFT. Sequence logo was generated by using weblogo (https://weblogo.berkeley.edu/logo.cgi).

**Reporting summary**. Further information on research design is available in the Nature Research Reporting Summary linked to this article.

## Data availability
The authors declare that the data supporting the findings of this study are available within the article and its Supplementary Information.

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

## Acknowledgements

This work is supported in part by grants from the National Key Research and Development Program (2018Y F A0900402 and 2021YFA0910501), and from the National Natural Science Foundation of China (21822703, 21921003, and 32070050), from the funding of Innovative research team of high-level local universities in Shanghai and a key laboratory program of the Education Commission of Shanghai Municipality (ZDSYS14005), and from West Light Foundation of The Chinese Academy of Sci-ences xbzg-zdsys-202105.

## Author contributions

Q.Z. and W.D. designed the project. S.G., S.W., S.M., and W.D. performed the research. S.G., S.W., S.M., Z.D., W.D., and Q.Z analyzed the data. S.G., S.W., W.D., and Q.Z. wrote the manuscript.

## Competing interests

The authors declare no competing interests.
