## [Peer Review File · Nature Communications]

REVIEWER COMMENTS

Reviewer #1 (Remarks to the Author):

In this paper the authors report on the enzymatic activity for DarE, a subclass of RiPP radical SAM enzyme that forms both ether and C-C crosslinks on three-residue motifs. The natural product resulting from DarE is darobactin, a new class of antibiotic with activity against gram-negative pathogens. The authors first demonstrate the activity on DarA by in vivo coexpression of NHis-DarA with DarE. The modified C-terminal fragment could be detected and was consistent with installation of ether and C-C crosslinks. The authors carry out invitro studies to show that DarE catalyzes the ether bond formation by dehydrogenation followed by hydration with water and -OH activation. While significantly more experimentation is needed, the results provided allow the authors to propose a reaction mechanism. The complete formation of the bicyclic ring system is proposed to use 3 mol eq of SAM. In the second part of the paper the authors conduct genome mining based on the DarE protein sequence using Blastp. A number of closely related DarE homologues and their precursors were identified. Functional studies were carried out on one closely related homologue and the product was characterized by NMR to show the same bicyclic crosslinks are formed. An additional precursor/enzyme pair was coexpressed leading to detection of only an ether bond formation. The amounts of this product were not sufficient to characterize by NMR. The proposed name daropeptides is given for aromatic-aliphatic ether containing peptides formed by this subclass of rSAM enzymes. The authors propose this as an additional source for antibiotics against gram-negative pathogens.

Overall this work provides the first functional study for DarE, a very important maturase responsible for the formation of a new class of antibiotics. The main results include functional validation of the rSAM enzyme in vivo and invitro. Most importantly, the authors provide evidence that the O involved in the ether linkage is supplied from water. A proposed mechanism is given for the remaining reactions but these will need to be explored further. In the remaining parts of the manuscript the authors characterize one additional darobactin analogue and evidence for an ether only forming rSAM enzyme in this class. Due to the relevance of the biosynthesis of this molecule and the first study to show in vitro activity for this enzyme this article is suitable for publication in Nature Communications. Some minor comments are made on some parts of the manuscript which should be addressed before being accepted.

Pg 4: Suggested to use Lytreptides instead of Streptide for this class (Comprehensive Natural Products III, 2.04)

Pg 5: Should DarApk be DarApk-1. 'decreases the proteolysis efficiency, allowing for the purification of the full length modified DarA pk in our analysis'

Pg 6/Figure 3: Since soluble NHis-DarE was obtained by coexpression of untagged DarA an additional control should be added for NHis-DarE digested with trypsin. This will show that modified DarA is not pulled down with NHis-DarE.

Pg 7: In the supporting information of the 18O experiment, the authors should show the comparison of the MSMS spectra (zoomed in) for labeled vs unlabeled to show that y1/y2 are identical and a clear difference is observed for y5 and beyond.

Pg 7: The authors should state whether a peak corresponding to the hydroxylated product (+16) could be observed.

Figure 1A: The planar chirality should be the same between the two rings (H. Kaur et al Nature). Therefore, the depiction of the W1 ring should be analogous to that of W3. It is suggested to orient the ring in W1 so the indole N is at the bottom similar to W3. The boxed representation in Figure S16C is a better depiction of W1. Also, fix the bond between WC α -NH so the bond is between the C and N.

Figure 1B: Give the scale for the genes. Indicate amino acid numbers on the precursor so the fragments described in the main text can be easily identified.

Supporting information

Pg 1: Change Bruker to Bruker

Pg 1: Where was the H218O obtained from and the purity?

Pg 5: Clarify what 10 mg is in the following sentence 'Reaction volume was typically 10 mg in 500 μ L and maintained at 37 °C for about 6 h'

Pg 5: What are the dimensions of this column... semi-preparative YMC-Triart C8 column

Figure S16: The symbols do not appear correctly in the boxes showing the J coupling..

Reviewer #2 (Remarks to the Author):

This manuscript by Guo et al. reported the biochemical and genetic characterization of a radical SAM enzyme (rSAM) in the biosynthesis of darobactin and other members of the daropeptide family. Darobactin was initially reported as a new type of RiPPs with an ether linkage and a C-C crosslink by the Lewis group in 2019 (Ref. 4) and possesses promising activities against Gram-negative bacteria. Its gene cluster comprising of five genes was reported in 2019 and two subsequent studies published in 2021 (Refs. 27 and 28) demonstrated that only one precursor peptide gene and one rSAM gene are required for the production of darobactin in heterologous hosts (e.g., *E. coli*). Furthermore, Ref 28 produced a series of mutated darobactin analogs through engineering the precursor peptide gene. In this work, the authors first observed the same results as Refs 27 and 28 that the rSAM (DarEpk) is catalytically active and sufficient to produce darobactin and modified precursor peptide containing darobactin when coexpressed with the His-tagged precursor peptide (DarApk) in *E. coli*. They further indicated that

endogenous proteases of *E. coli* cleave the leader and follower peptide of DarA. Subsequently, the authors prepared recombinant DarEpK and confirmed its catalytic functions in vitro. Significantly, the biochemical studies suggested that the ether bond formation between Trp49 and Trp 51 is the first biosynthetic step and the ether oxygen comes from water, leading to the development of a plausible biosynthetic pathway. Finally, the authors identified additional DarE homologs that were classified into four clusters (darobactin belongs to Cluster I). They then heterologously produced the peptides of Cluster III and IV, and determined the structure of the one of Cluster III. Of note, the product of Cluster IV carried only the ether crosslinkage.

Overall, this work uncovered new chemistry of rSAM enzymes in the RiPPs biosynthesis. Of note, the ether crosslinkage formed through dehydrogenation and hydration reactions is highly novel. Furthermore, the identification of new darobactin analogs is another significant result. On the other hand, the work is considered to be relatively under developed as shown below.

- (1). The product of Cluster IV carried only the ether crosslinkage, supporting the ether formation as the first biosynthetic step of Cluster I. On the other hand, it is recommended to perform additional studies to validate it (e.g., mutagenesis of K5).
- (2). Additional studies are recommended to investigate if W1 and W3 are indispensable for the formation of the ether and C-C linkages.
- (3). Additional studies are recommended to investigate if rSAM enzymes of different clusters are promiscuous toward precursor peptides of other clusters.
- (4). Bioactivity tests of cluster III and IV.

Minor:

- (1). Calculate the ratio of O18 incorporated product (Figure 3D). Also include calculated and observed MWs.
- (2). Figure S7: was DarA copurified? Copurified DarA may affect the results of DarEpk biochemical studies.

REVIEWER COMMENTS

Reviewer #1 (Remarks to the Author):

In this paper the authors report on the enzymatic activity for DarE, a subclass of RiPP radical SAM enzyme that forms both ether and C-C crosslinks on three-residue motifs. The natural product resulting from DarE is darobactin, a new class of antibiotic with activity against gram-negative pathogens. The authors first demonstrate the activity on DarA by in vivo coexpression of NHis-DarA with DarE. The modified C-terminal fragment could be detected and was consistent with installation of ether and C-C crosslinks. The authors carry out invitro studies to show that DarE catalyzes the ether bond formation by dehydrogenation followed by hydration with water and -OH activation. While significantly more experimentation is needed, the results provided allow the authors to propose a reaction mechanism. The complete formation of the bicyclic ring system is proposed to use 3 mol eq of SAM. In the second part of the paper the authors conduct genome mining based on the DarE protein sequence using Blastp. A number of closely related DarE homologues and their precursors were identified. Functional studies were carried out one closely related homologue and the product was characterized by NMR to show the same bicyclic crosslinks are formed. An additional precursor/enzyme pair was coexpressed leading to detection of only an ether bond formation. The amounts of this product were not sufficient to characterize by NMR. The proposed name daropeptides is given for aromatic-aliphatic ether containing peptides formed by this subclass of rSAM enzymes. The authors propose this as an additional source for antibiotics against gram-negative pathogens.

Overall this work provides the first functional study for DarE, a very important maturase responsible for the formation of a new class of antibiotics. The main results include functional validation of the rSAM enzyme in vivo and invitro. Most importantly, the authors provide evidence that the O involved in the ether linkage is supplied from water. A proposed mechanism is given for the remaining reactions but these will need to be explored further. In the remaining parts of the manuscript the authors characterize one additional darobactin analogue and evidence for an ether only forming rSAM enzyme in this class. Due to the relevance of the biosynthesis of this molecule and the first study to show in vitro activity for this enzyme this article is suitable for publication in Nature Communications. Some minor comments are made on some parts of the manuscript which should be addressed before being accepted.

We thank the reviewer for the recognition of our paper.

Pg 4: Suggested to use Lytreptides instead of Streptide for this class (Comprehensive Natural Products III, 2.04)

Revision was made according to the suggestion.

Pg 5: Should DarApk be DarApk-1. ‘decreases the proteolysis efficiency, allowing for the purification of the full length modified DarA pk in our analysis’

DarApk was corrected to DarApk-1 in the revised manuscript.

Pg 6/Figure 3: Since soluble NHis-DarE was obtained by coexpression of untagged DarA an additional control should be added for NHis-DarE digested with trypsin. This will show that modified DarA is not pulled down with NHis-DarE.

We thank the reviewer for this suggestion. The control assay without the precursor peptide substrate (i.e. DarE with SAM and other components) was added to Figure 3 in the revised manuscript.

Pg 7: In the supporting information of the 180 experiment, the authors should show the comparison of the MSMS spectra (zoomed in) for labeled vs unlabeled to show that y1/y2 are identical and a clear difference is observed for y5 and beyond.

We thank the reviewer for the suggestion. Detailed comparison of labeled vs unlabeled zoom-in MSMS spectra is now provided in Figure S12 in the revised SI.

Pg 7: The authors should state whether a peak corresponding to the hydroxylated product (+16) could be observed.

We did not observe the hydroxylated product in the assay with wild type DarApk. However, the hydroxylated product was indeed observed for the W49F mutant of DarApk, which is consistent with our mechanistic proposal shown in Figure 4. The new data was provided in the revised manuscript and SI (e.g. Figure S23).

Figure 1A: The planar chirality should be the same between the two rings (H. Kaur et al Nature). Therefore, the depiction of the W1 ring should be analogous to that of W3. It is suggested to orient the ring in W1 so the indole N is at the bottom similar to W3. The boxed representation in Figure S16C is a better depiction of W1. Also, fix the bond between WC α -NH so the bond is between the C and N.

The structure in Figure 1A and Figure S16C were revised according to the reviewer's suggestion.

Figure 1B: Give the scale for the genes. Indicate amino acid numbers on the precursor so the fragments described in the main text can be easily identified. The scale for the gene cluster in Figure 1B and the amino acid numbers were provided in the revised Figure 1B.

Supporting information

Pg 1: Change Brucker to Bruker

Change was made.

Pg 1: Where was the H2180 obtained from and the purity?

The commercial source and purity of H2180 was provided in the revised SI.

Clarify what 10 mg is in the following sentence ‘Reaction volume was typically 10 mg in 500 L and maintained at 37 ° C for about 6 h’

Change was made.

What are the dimensions of this column... semi-preparative YMC-Triart C8 column

The dimensions of the column is provided in the revised SI.

Figure S16: The symbols do not appear correctly in the boxes showing the J coupling..

The symbols in Figure S16 was revised.

Reviewer #2 (Remarks to the Author):

This manuscript by Guo et al. reported the biochemical and genetic characterization of a radical SAM enzyme (rSAM) in the biosynthesis of darobactin and other members of the daropeptide family. Darobactin was initially reported as a new type of RiPPs with an ether linkage and a C-C crosslink by the Lewis group in 2019 (Ref. 4) and possesses promising activities against Gram-negative bacteria. Its gene cluster comprising of five genes was reported in 2019 and two subsequent studies published in 2021 (Refs. 27 and 28) demonstrated that only one precursor peptide gene and one rSAM gene are required for the production of darobactin in heterologous hosts (e.g., *E. coli*). Furthermore, Ref 28 produced a series of mutated darobactin analogs through engineering the precursor peptide gene. In this work, the authors first observed the same results as Refs 27 and 28 that the rSAM (DarEpk) is catalytically active and sufficient to produce darobactin and modified precursor peptide containing darobactin when coexpressed with the His-tagged precursor peptide (DarApk) in *E. coli*. They further indicated that endogenous proteases of *E. coli* cleave the leader and follower peptide of DarA. Subsequently, the authors prepared recombinant DarEpK and confirmed its catalytic functions in vitro. Significantly, the biochemical studies suggested that the ether bond formation between Trp49 and Trp 51 is the first biosynthetic step and the ether oxygen comes from water, leading to the development of a plausible biosynthetic pathway. Finally, the authors identified additional DarE homologs that were classified into four clusters (darobactin belongs to Cluster I). They then heterologously produced the peptides of Cluster III and IV, and determined the structure of the one of Cluster III. Of note, the product of Cluster IV carried only the ether crosslinkage.

Overall, this work uncovered new chemistry of rSAM enzymes in the RiPPs biosynthesis. Of note, the ether crosslinkage formed through dehydrogenation and hydration reactions is highly novel. Furthermore, the identification of new darobactin analogs is another significant result. On the other hand, the work is considered to be relatively under developed as shown below.

We thank the reviewer for the recognition of the novelty of our work. Additional experiments were provided in the revised manuscript.

(1). The product of Cluster IV carried only the ether crosslinkage, supporting

the ether formation as the first biosynthetic step of Cluster I. On the other hand, it is recommended to perform additional studies to validate it (e.g., mutagenesis of K5).

We thank the reviewer for the suggestion. We changed the K5 in DarApk to Ala, and the result showed that the resulting DarA_{pk} mutant mainly harbored a +14 modification, which occurred between Trp48 and Trp51 (Figure S9 in the revised SI). This result is consistent with the proposal that ether bond formation precedes the C-C bond formation in darobactin biosynthesis. These results are added to the revised manuscript and SI.

(2). Additional studies are recommended to investigate if W1 and W3 are indispensable for the formation of the ether and C-C linkages.

We thank the reviewer for the suggestion. We changed W3 (i.e. Trp51) to Phe, and the result found that the ether-crosslink was not found in the Trp-to-Phe mutant DarA. We also changed W1 (i.e. Trp49) to Phe. However, we only observed hydroxylation of W3 (i.e. Trp51) whereas the expected ether-crosslink was not formed in the mutant DarA. These results were incorporated into the revised manuscript and SI (e.g. Figure S22-S23).

(3). Additional studies are recommended to investigate if rSAM enzymes of different clusters are promiscuous toward precursor peptides of other clusters. We thank the reviewer for the suggestion. We coexpressed DarApk with DarEyi, and DarAyi with DarApk, showing that although DarA can be modified by DarE enzymes from different clusters, these enzymes are likely not as efficient as the native enzyme. Observation of the monocyclic ether-crosslinked DarA_{pk} in these analysis again supported that ether-crosslink precedes the C-C bond formation in darobactin biosynthesis. These results were incorporated into the revised manuscript and SI (e.g. Figure S24-S25).

(4). Bioactivity tests of cluster III and IV.

Despite the difference, the precursor peptides from cluster III generally have similar core peptides with those from cluster I, and the very recent studies by Muller and coworker (ref. 28) showed that the mature peptide from DarAyi has similar activity with darobactin. Further studies on the bioactivity of product from cluster IV is currently underway in our lab.

Minor:

(1). Calculate the ratio of 018 incorporated product (Figure 3D). Also include calculated and observed MWs.

The ratio of 180 incorporation as well as the molecular weight of the 180-labeled product were now provided in the revised Figure 3D and the figure caption.

(2). Figure S7: was DarA copurified? Copurified DarA may affect the results of DarEpk biochemical studies.

As also mentioned early, the control assay with the DarE enzyme alone were provided in the revised Figure 3, showing that untagged DarA was not an issue.

REVIEWERS' COMMENTS

Reviewer #1 (Remarks to the Author):

The revised manuscript addressed the comments of this reviewer, has improved the clarity, and the additional experiments have improved the quality of the manuscript. It is now suitable for publication.

Reviewer #2 (Remarks to the Author):

This revised manuscript properly addressed comments raised previously and is suitable for the publication.